# Minimizing Structural Vibrations via Guided Diffusion Design Optimization

**Jan van Delden**[1], **Julius Schultz**[2], **Christopher Blech**[2], **Sabine C. Langer**[2], **Timo Lüddecke**[1]

[1]Institute of Computer Science, University Göttingen
[2]Institute for Acoustics and Dynamics, TU Braunschweig
Correspondence: `jan.vandelden@uni-goettingen.de`

## Abstract

Structural vibrations are a source of unwanted noise in everyday products like cars, trains or airplanes. For example, the motor of a car causes the chassis to vibrate, which radiates sound into the interior of the car and is eventually perceived by the passenger as noise. Because of this, engineers try to minimize the amount of vibration to reduce noise. This work introduces a method for reducing vibrations by optimally placing beadings (indentations) in plate-like structures with a guided diffusion design optimization approach. Our approach integrates a diffusion model for realistic design generation and the gradient information from a surrogate model trained to predict the vibration patterns of a design to guide the design towards low-vibration energy. Results demonstrate that our method generates plates with lower vibration energy than any sample within the training dataset. To enhance broader applicability, further development is needed in incorporating constraints in the outcome plate design. Code and example notebook: `https://github.com/ecker-lab/diffusion_minimizing_vibrations`

## 1 Introduction

Mechanical structures emit unwanted noise in everyday situations, which causes discomfort and health problems (Basner et al., 2014). To address this, engineers work on reducing the noise emitted by mechanical structures. In this work, we approach this issue by focusing on reducing structural vibrations. Vibrating structures cause the surrounding air to vibrate, producing sound. In vehicles or machinery, vibrations transmit sound from a source like engines to passenger areas.

How can engineers reduce structural vibrations? Applying damping material is effective, but adds weight and bulk. Altering the form or material of the design of a mechanical may compromise its primary function. To address this, we focus on beadings: local indentations in plate-like structures that increase the stiffness. We consider the design optimization task of placing beadings on plates of a fixed size based on a benchmark dataset (van Delden et al., 2023).

How can an engineer tackle the optimization task? Sampling a large number of design permutations and numerically solving the system via e.g. the finite element method could identify the 'most quiet' within the tested designs. This brute-force approach quickly becomes computationally prohibitive in large design spaces. Alternatively, there exist methods relying on numerically calculated physical quantities, like the shear stress or strains, that place beadings based on this information. This approach offers some vibration reduction but is limited in its effectiveness (Rothe, 2022).

In this work, we present a generative deep learning method for vibration reduction. Our approach combines input space optimization (e.g. Gatys et al., 2015; Olah et al., 2017) with diffusion models (Sohl-Dickstein et al., 2015; Ho et al., 2020) to arrive at a method similar to classifier guided diffusion (Dhariwal & Nichol, 2021). Iteratively refining plate geometries within this framework, we achieve designs with vibration velocities well below any plates in our training dataset. This outcome highlights the potential of generative deep learning methods for design optimization in acoustics.

## 2 DATA AND PHYSICAL MODEL

In this work, we adopt the physical model and dataset in the V-5000 setting introduced by van Delden et al. (2023). This dataset focuses on the vibrational behavior of rectangular plates in response to an external force. The plates are modeled based on an established differential equation and the vibrational behavior is numerically computed. In the following, we provide a technical description of the physical model and dataset.

The considered physical model is a simply supported rectangular plate, excited by a point force with a harmonic excitation frequency. We are interested in the dynamic response of the plate in the frequency domain. The plate theory by Mindlin is employed, which is a valid differential equation for moderately thin plates (Mindlin, 1951). A shell formulation is derived which combines the plate theory with a disc formulation for in-plane loads and enables arbitrarily formed moderately thin structures. The Finite Element Method (FEM) is applied to numerically solve the differential equation. To this end, the plate is discretized by a regular grid of 121 x 81 nodes and meshed with triangular shell elements using linear ansatz functions. The choice of triangular elements is a robust approach to incorporate arbitrary beading patterns. After discretization, the system matrices are assembled and the discrete dynamical system in the frequency domain is obtained as

$$(-\Omega^2 \mathbf{M} + \mathbf{K}(\mathbf{g}))\mathbf{u} = \mathbf{f}.$$

Here $\Omega$, denotes the excitation frequency, $\mathbf{M}$, $\mathbf{K}$ the mass and stiffness matrix and $\mathbf{f}$ and $\mathbf{u}$ the load and solution vector. The stiffness matrix $\mathbf{K}$ depends on the beading pattern geometry $\mathbf{g}$. The important quantity for sound radiation is the velocity component orthogonal to the plate (z-direction). By $\mathbf{u}_z \subset \mathbf{u}$ we denote the z-displacement, which is a subset of all degrees of freedom $\mathbf{u}$. The z-velocity is obtained by $\mathbf{v}_z = i\Omega\mathbf{u}_z$. To arrive at a more compact spatially averaged quantity, we consider the absolute mean squared velocity:

$$\bar{v}_z = \frac{1}{n} \sum_{j=1}^{n} |v_z^{(j)}|^2$$

As this quantity is proportional to the kinetic energy, it is closely related to how much sound is radiated. The quantity is represented in a dB scale and therefore log-transformed according to van Delden et al. (2023). It is represented as a function of the frequency $\Omega \mapsto \bar{v}_z(\Omega)$. The frequency averaged mean squared velocity $v_\Sigma$ defines an integral quantity for a frequency interval by

$$v_\Sigma = \frac{1}{\Omega_1 - \Omega_0} \int_{\Omega_0}^{\Omega_1} \bar{v}_z(\Omega) d\Omega$$

In the dataset, 6000 samples for training and evaluation are available. Each sample includes a beading pattern $\mathbf{g}^{(i)}$, the corresponding velocity fields $\mathbf{v}_z^{(i)}$ and the discretized frequency response function $\bar{\mathbf{v}}^{(i)}$. The beading patterns consist of up to three randomly placed lines and up to two ellipses. Solutions for excitation frequencies from 1 - 300 Hz are included.

## 3 GUIDED DIFFUSION FOR DESIGN OPTIMIZATION

Gradient-based input-space optimization methods operate by defining a loss function and performing gradient-descent on the input to the neural network based on the loss (e.g. Gatys et al., 2015; Olah et al., 2017). This method was for example employed for image style transfer and generating images that lead to maximum activation of outputs of the neural network. The resulting images often diverge from the training data distribution the neural network was trained on, since no mechanism enforces faithfulness. For design tasks in engineering, more control over the outcome is necessary.

Generative modeling methods that model a data distribution address this issue. They allow for generating new samples from a distribution that can be defined by a training dataset. Recently, the closely related score-based and denoising-diffusion based generative models have been prominent (Sohl-Dickstein et al., 2015; Ho et al., 2020). In those methods, a neural network is trained to estimate the score function of a distribution $\mathbf{X}$ defined by a dataset (Song et al., 2020). The score function $\mathbf{s}$ of a distribution is defined as $\mathbf{s}(x) = \nabla_x log(p(x))$. By iteratively updating a random initial input $x_0$ with $x_{i+1} = x_i + \mathbf{s}(x_i)$ for a large number of steps, new samples from $\mathbf{X}$ can be generated.

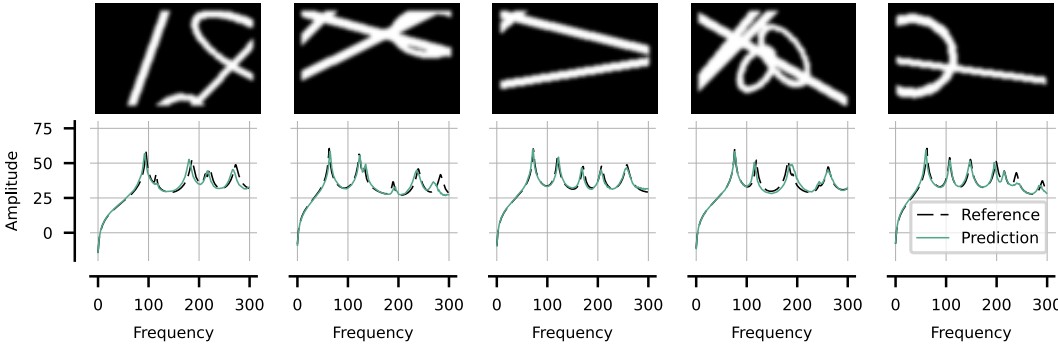

Figure 1: Randomly selected plate geometries along with their frequency response and the predictions from our regression model.

Diffusion models can be combined with guidance strategies, that enable sampling from conditional probability distributions $P(X|C)$ such as classifier-guidance (Dhariwal & Nichol, 2021), or classifier-free-guidance (Ho & Salimans, 2022). Classifier guidance adds a second term to the iterative update step:

$$x_{i+1} = x_i + \alpha_i \mathbf{s}(x_i) + \beta_i \mathbf{s}(c|x_i)$$

$\mathbf{s}(x_i)$ is predicted by a diffusion model. $\mathbf{s}(c|x_i)$ is the gradient in $x_i$ computed through backpropagation from a pretrained classifier. Additional scaling parameters $\alpha$ and $\beta$ allow controlling the influence of the respective terms. By selecting an appropriate condition $c$, samples with desired properties can be generated. To apply this principle to minimizing vibrations, we have to replace the $\mathbf{s}(c|x_i)$ by a loss function, since minimal noise is not a discrete condition. This approach of replacing the score function by a more general function has been applied in previous work (e.g. Pierzchlewicz et al., 2023; Mazé & Ahmed, 2023) and described as either regressor-guidance or energy-guidance.

## 3.1 FRAMEWORK

For generating plates with optimized vibrational properties, we adopt the previously described denoising diffusion generative modeling paradigm and train two neural networks. One denoising diffusion generative model $\theta$ is trained to generate beading patterns as 2d images. $\theta$ is based on a standard UNet architecture with 10 Mio. weights and trained for 10000 iterations on batches of 512 continuously newly sampled beading patterns as described in Section 2. 500 denoising steps are set.

A second neural network $\eta$ is trained to predict the velocity fields $\mathbf{v}_z$ and frequency response function $\bar{\mathbf{v}}_z$ given a plate geometry $\mathbf{g}$ and is called regression model in the following. The UNet network architecture and training settings from van Delden et al. (2023) are adopted. The UNet has 7 Mio. weights and takes as input a frequency query together with the plate geometry. Example predictions are shown in Figure 1. Accurate modeling of the score function even for noisy plate designs is required (Song et al., 2020). Because of this, we train the regression model with noised plate geometries by adding random pixelwise noise. The noise is sampled from a standard normal distribution scaled by 50 % of the input pixel range. Pixel values are clamped to stay in the original value range.

**Guided Diffusion.** To perform guided diffusion, we employ both the already trained diffusion model $\theta$ and the already trained regression model $\eta$. To perform one denoising step, we employ the DDPM scheduler function $d(\cdot)$ from Ho et al. (2020), which in each step injects new noise in addition to denoising the input. For $\eta$, $\mathbf{L}(\eta(x))$ is defined, a loss function that sums up the frequency response prediction given a plate geometry $x$. Then, the gradient of the loss function with respect to the plate geometry $\nabla_x \mathbf{L}(\eta(x))$ is obtained. As the diffusion model $\theta$ operates on Gaussian noise and the regression model $\eta$ operates on images in the range [0,1], $x$ is converted to the correct scale and squared before being passed to $\eta$, which empirically enhances results, possibly by aligning the noisy plates closer to the training data distribution. To arrive at a new plate design, noise is sampled from a normal distribution as $x_0$ and then the following update step is iteratively performed:

$$x_{i+1} = d(x_i, \alpha_i, \theta(x_i)) - \beta_i \nabla_{x_i} \mathbf{L}(\eta(x_i))$$

In total 500 update steps are performed to match the training of the diffusion model. The denoising parameter $\alpha$ is set to a cosine schedule with an initial value of 0.02 and a final value of 0.0001. $\beta$ is

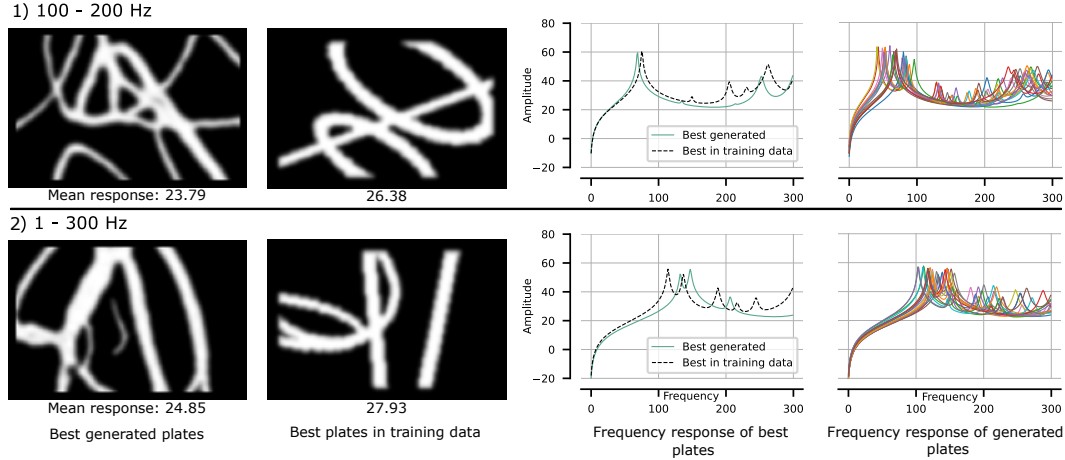

Figure 2: Exemplary generation results with lowest mean response out of 64 generations (left). First row is optimized for low responses between 100 - 200 Hz, second row for 1 - 300 Hz. Responses in dB scale. Further visualizations in Appendix.

set to a cosine schedule with an initial value of $0.01$ and a final value of $0.0001$. In consequence, the changes are larger for small $i$, when the geometry still contains a lot of noise and become smaller towards the end of the generation process. These settings proved to generate new plate geometries that remain close to the training data distribution of the diffusion model and could in principle be manufactured.

## 4 EXPERIMENTS

To assess the capabilities of the method to generate plate geometries with desired vibrational properties, we employ the diffusion model in conjunction with the regression model and alternate between performing one denoising diffusion and one gradient descent step. We design two experiments: (Targeted, 1) The loss is set to the sum of the frequency response for the frequency range 100 - 200 Hz. This setting of reducing the vibrations for a targeted range of excitation frequencies is common in engineering design. (Full range, 2) The loss is set to the sum of the predicted frequency response for all frequencies (1 - 300 Hz). After having performed diverse experiments with the second setting (full range, 2), 415 additional plates along with numerical solutions were available. These generated plates are more diverse than the plates available in the original training dataset and were used to fine-tune the regression model to achieve better prediction accuracy for more diverse plate geometries. We report generated results with the base model as well as with the fine-tuned model and compute 32 plates per condition. The frequency response of the generated designs was obtained via numerical simulation.

For both loss conditions, our method is able to produce better results than any sample within the 5000 training data points (Figure 2, further generated plates in Appendix). The generated plate geometries look distinctly different to the training data and have more variety in their shapes. Comparing the two loss conditions, the produced patterns for the targeted setting (100 - 200 Hz) exhibit less thick beadings and in total less beaded area. The produced patterns for the full range setting (1 - 300 Hz) contain distinct thick beadings from top to bottom. The associated frequency responses differ, especially in the positioning of the first eigenfrequency. For the full range setting, the first eigenfrequency is shifted upwards to around 150 Hz for many plates. A second possible position of the first eigenfrequency seems to be around 110 Hz. In contrast, for the targeted setting the first eigenfrequency is shifted downwards well below 100 Hz.

Due to the differences between the generated plates and the plates within the training data, the predicted responses from the regression model differ more strongly from the numerically computed responses than test results for the regression model would suggest (see Table 1, visualizations in Appendix). Fine-tuning with in-domain plates generated from our method reduced the prediction

error slightly for the full range, but lead to worse predictions for the targeted setting. A reason for this could be that the additional plates were mainly generated for the full range setting.

Table 1: Mean frequency responses in dB from 32 generated plates per condition. Values in parentheses are predicted mean frequency responses from the regression model.

| Optimization range | Base model | Fine-tuned model |
|---|---|---|
| 100 - 200 Hz | 27.5 (26.8) | 30.3 (28.2) |
| 1 - 300 Hz | 28.0 (26.7) | 26.8 (26.0) |

## 5 CONCLUSION

In this work, we presented a guided diffusion based method to generate novel plate designs with optimized vibrational properties. Our method is able to generate plate designs with a lower response than any plate within the training data for two different frequency ranges. This could help in reducing noise produced by mechanical structures. The plate designs are novel compared to the training data, but do not deviate very far from the design space, which balances between novelty of the designs and actual manufacturability of the plates. The resulting beading patterns differ substantially based on the optimization target.

**Limitations.** To directly apply this method to design tasks, several limitations still need to be addressed. The efficiency of the method is questionable, since it requires a gradient-based regression model that is usually trained with previously simulated training data for a specific design task. Including additional constraints into the design task would be necessary, e.g. the placement of regions where no beadings may be placed. Better convergence of the gradient based optimization or alternatively more flexibility in the number of optimization steps would be beneficial.

**Acknowledgements.** This research is funded by the Deutsche Forschungsgemeinschaft (DFG, German Research Foundation), Project number 501927736, within the DFG Priority Programme 'SPP 2353: Daring More Intelligence - Design Assistants in Mechanics and Dynamics'. This support is highly appreciated.

## REPRODUCIBILITY STATEMENT

The training data used in this work is publically available from a dataset repository as described in van Delden et al. (2023). All deep learning model can be trained on a single A100 GPU in a day at maximum. A new design can be generated in less than a minute. The code and trained models are available from `https://github.com/ecker-lab/diffusion_minimizing_vibrations`. To numerically compute the vibration patterns, commercially available simulation software is necessary.

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

## A APPENDIX

### A.1 RESULTS FOR FULL FREQUENCY RANGE - BASE MODEL

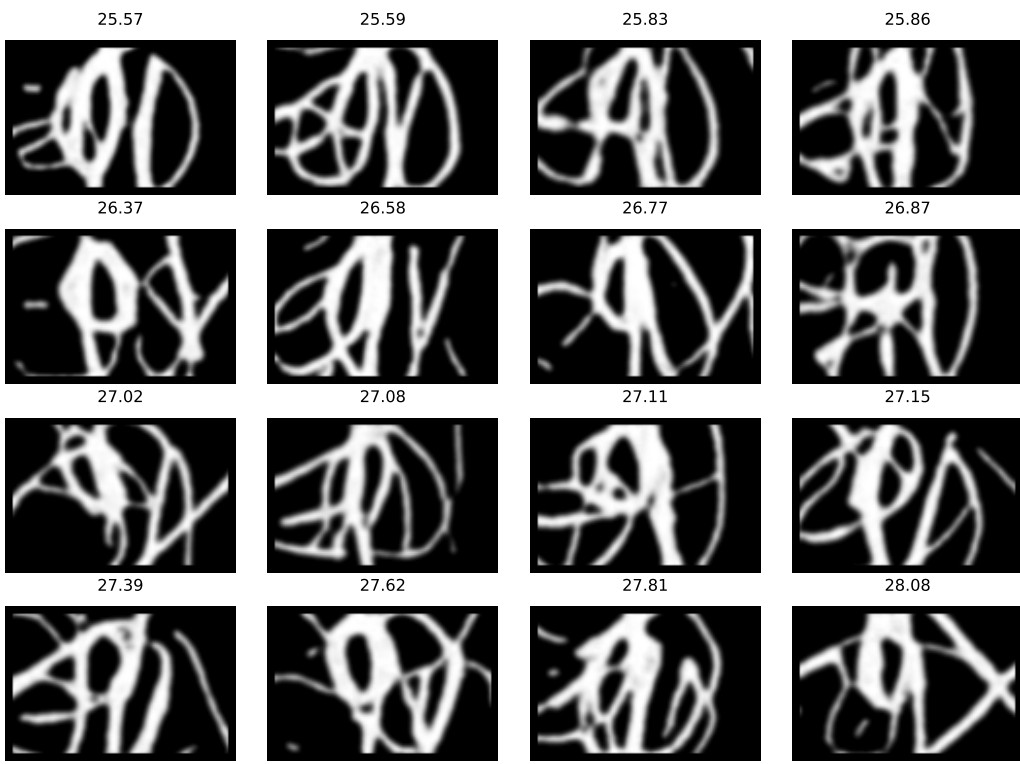

Figure 3: 16 best generation results out of 32. Optimization target: Minimize frequency response in full frequency range. Base model. The number above the plate designs gives the mean frequency response.

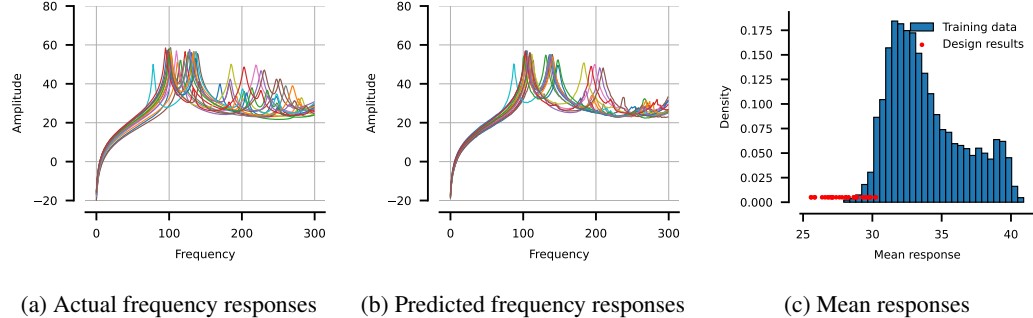

(a) Actual frequency responses     (b) Predicted frequency responses     (c) Mean responses

Figure 4: Statistics of the generated plates. (a) shows the actual frequency response function as computed by the numerical solver. (b) shows the predicted frequency response function from the regression model. (a) and (b) show the 16 best generated plates. (c) shows the achieved mean responses in comparison to the distribution in the training dataset for all 32 plates.

## A.2 RESULTS FOR FULL FREQUENCY RANGE - FINE-TUNED MODEL

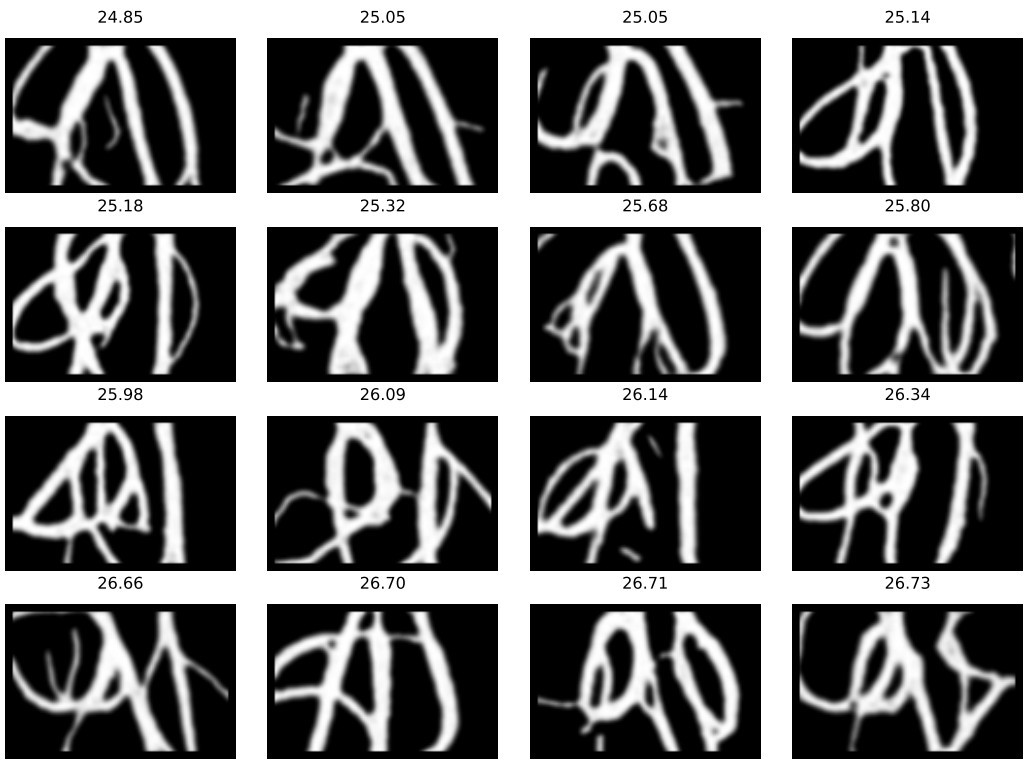

Figure 5: 16 best generation results out of 32. Optimization target: Minimize frequency response in full frequency range. Fine-tuned model. The number above the plate designs gives the mean frequency response.

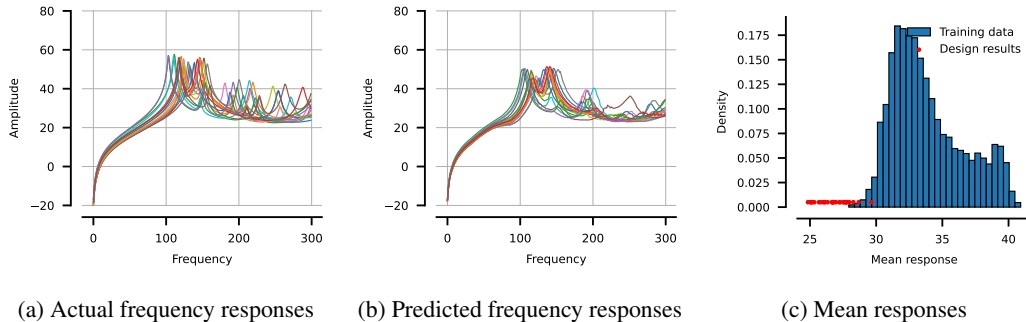

(a) Actual frequency responses     (b) Predicted frequency responses     (c) Mean responses

Figure 6: Statistics of the generated plates. (a) shows the actual frequency response function as computed by the numerical solver. (b) shows the predicted frequency response function from the regression model. (a) and (b) show the 16 best generated plates. (c) shows the achieved mean responses in comparison to the distribution in the training dataset for all 32 plates.

## A.3    RESULTS FOR 100 - 200 HZ - BASE MODEL

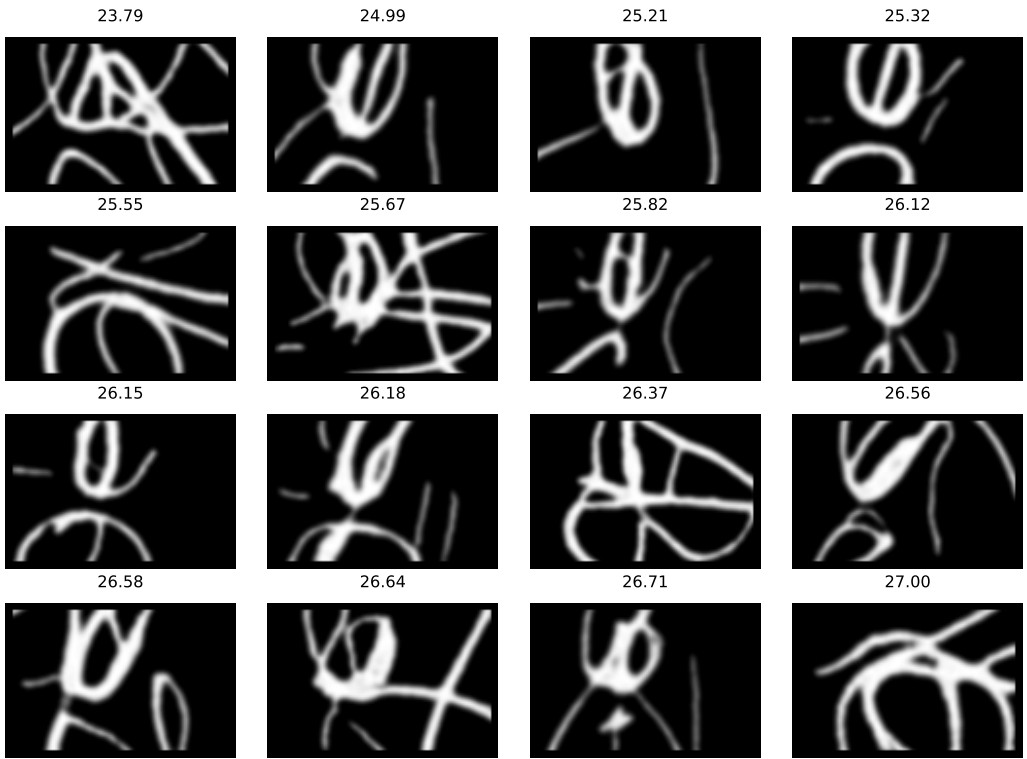

Figure 7: 16 best generation results out of 32. Optimization target: Minimize full frequency range. Base model. The number above the plate designs gives the mean frequency response.

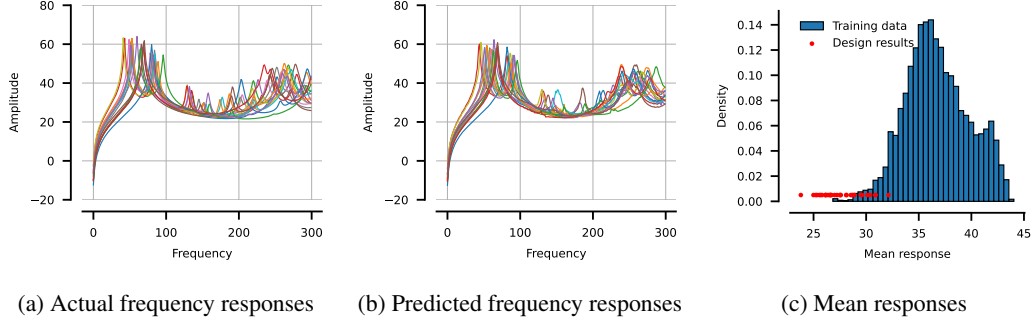

(a) Actual frequency responses        (b) Predicted frequency responses        (c) Mean responses

Figure 8: Statistics of the generated plates. (a) shows the actual frequency response function as computed by the numerical solver. (b) shows the predicted frequency response function from the regression model. (a) and (b) show the 16 best generated plates. (c) shows the achieved mean responses in comparison to the distribution in the training dataset for all 32 plates.

A.4 RESULTS FOR 100 - 200 HZ - FINE-TUNED MODEL

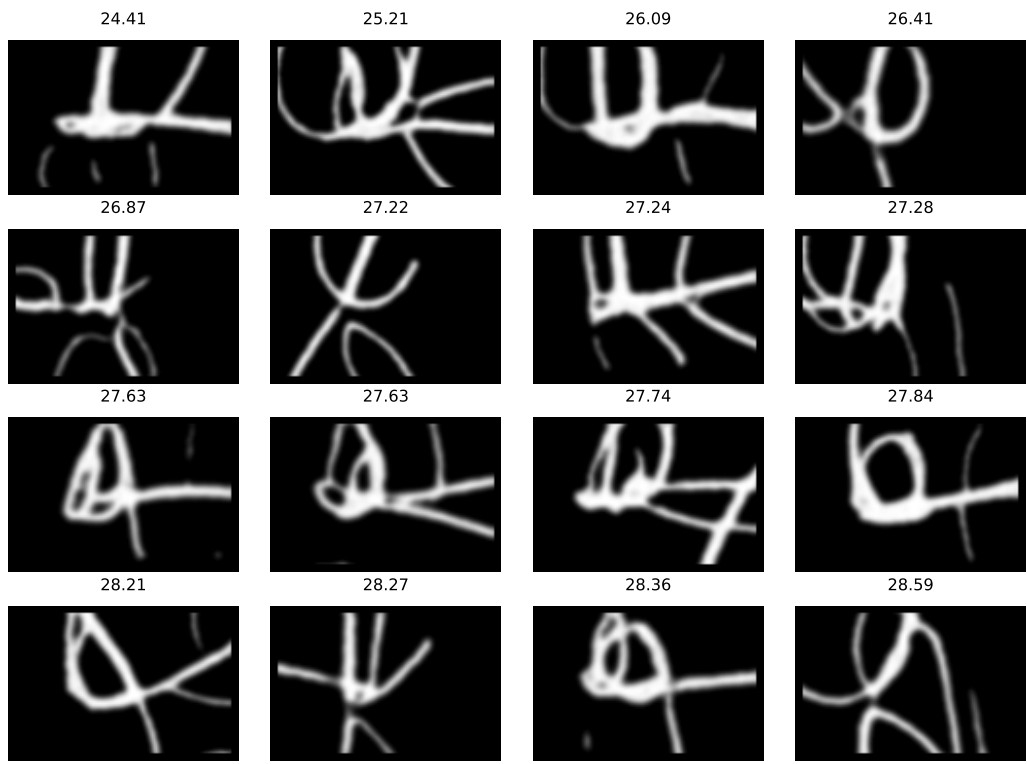

Figure 9: 16 best generation results out of 32. Optimization target: Minimize full frequency range. Fine-tuned model. The number above the plate designs gives the mean frequency response.

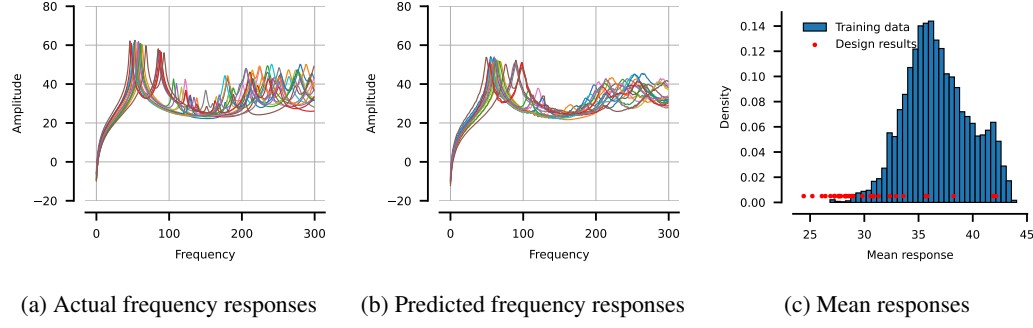

(a) Actual frequency responses  (b) Predicted frequency responses  (c) Mean responses

Figure 10: Statistics of the generated plates. (a) shows the actual frequency response function as computed by the numerical solver. (b) shows the predicted frequency response function from the regression model. (a) and (b) show the 16 best generated plates. (c) shows the achieved mean responses in comparison to the distribution in the training dataset for all 32 plates.

