# OpenReview forum: "Minimizing Structural Vibrations via Guided Diffusion Design Optimization"
_ICLR.cc/2024/Workshop/AI4DiffEqtnsInSci — AI4DiffEqtnsInSci @ ICLR 2024 Poster_

### Official Review · Reviewer_mhvF · 2024-02-24
**Innovative but needs further clarification**

**Rating:** 7
**Confidence:** 3

**Review:**

The paper proposes a novel generative DL approach to reduce structural vibrations in plate-like structures using a guided diffusion design optimization framework. By integrating a denoising diffusion generative model with a surrogate model for vibration prediction, the authors propose a method that generates design patterns with significantly reduced vibration energy.

Pros:
- The application of guided diffusion models for structural design optimization in traditional engineering is innovative.

Cons:
- While the audience may be interested, the paper's contribution may not be directly relevant to the workshop topic. It seems that the differential equations are only used to generate datasets, rather than directly guiding or benefiting from the model.

Questions/Comments:
- May need to justify the beading patterns (i.e., up to 3 lines and 2 ellipses) in the dataset and the applicability of the approach to other scenarios (e.g., possibly more complex patterns in realistic).
- The methodological steps involved in combining these models for design optimization are not sufficiently detailed to be easily understood by a reader unfamiliar with these techniques. For example, why are "score-based and denoising-diffusion based" generative models used? How alpha and beta were determined and adjusted in the guidance?
-  The advantages and limitations of the method are not clear, as there is no detailed analysis comparing the proposed approach with traditional design optimization techniques.
-  At least pseudocode or a more detailed description of the algorithm within the paper would allow for a better preliminary evaluation.

---

### Official Review · Reviewer_Gfyd · 2024-02-27
**Review of GDDO paper**

**Rating:** 7
**Confidence:** 3

**Review:**

This is a review of the manuscript, "Minimizing structural vibrations via guided diffusion design optimization," submitted to the ICLR 2024 Workshop on AI4DifferentialEquations In Science}. The paper describes a diffusion-model-based method to design a beading pattern for vibration control (damping). The authors report a modified thin-plate system that responds to external forcing, and the response is characterized via the frequency-integrated mean-square velocity. Two neural networks, $\theta$ and $\eta$, are trained to generate beading patterns and predict the frequency response for a specified plate geometry, respectively. In both cases, a U-Net architecture is chosen. A guided-diffusion algorithm utilizes $\theta$ and $\eta$ to iteratively update a plate design, starting from a random seed. Numerical experiments that demonstrate the ability of this approach to generate designs which are superior to all elements of the training dataset are reported.\par

The paper is well-written and provides a helpful overview of the physical problem, and the work is of a high quality. Moreover, the authors note the primary limitation of their technique, namely: the high-cost and parochial scope of the trained networks.

---

### Meta-Review · Area_Chair_j2kv · 2024-03-02

**Recommendation:** Accept (Poster)

**Metareview:**

The paper proposes a novel generative DL approach to reduce structural vibrations in plate-like structures using a guided diffusion design optimization framework. By integrating a de-noising diffusion generative model with a surrogate model for vibration prediction, the authors propose a method that generates design patterns with significantly reduced vibration energy. The work is very interesting but has limitations in not being compared to traditional approaches and there is limited more complex examples to prove it's generalizability. For these reasons it should be accepted for the poster session.

---

### Decision · Program_Chairs · 2024-03-02

Accept (Poster)